# Telomeric double-strand DNA-binding proteins DTN-1 and DTN-2 ensure germline immortality in *Caenorhabditis elegans*

Io Yamamoto[1], Kexin Zhang[1], Jingjing Zhang[1], Egor Vorontsov[2], Hiroki Shibuya[1]*

[1]Department of Chemistry and Molecular Biology, University of Gothenburg, Gothenburg, Sweden; [2]Proteomics Core Facility, Sahlgrenska Academy, University of Gothenburg, Gothenburg, Sweden

**Abstract** Telomeres are nucleoprotein complexes at the ends of chromosomes and are indispensable for the protection and lengthening of terminal DNA. Despite the evolutionarily conserved roles of telomeres, the telomeric double-strand DNA (dsDNA)-binding proteins have evolved rapidly. Here, we identified double-strand telomeric DNA-binding proteins (DTN-1 and DTN-2) in *Caenorhabditis elegans* as non-canonical telomeric dsDNA-binding proteins. DTN-1 and DTN-2 are paralogous proteins that have three putative MYB-like DNA-binding domains and bind to telomeric dsDNA in a sequence-specific manner. DTN-1 and DTN-2 form complexes with the single-strand telomeric DNA-binding proteins POT-1 and POT-2 and constitutively localize to telomeres. The *dtn-1* and *dtn-2* genes function redundantly, and their simultaneous deletion results in progressive germline mortality, which accompanies telomere hyper-elongation and chromosomal bridges. Our study suggests that DTN-1 and DTN-2 are core shelterin components in *C. elegans* telomeres that act as negative regulators of telomere length and are essential for germline immortality.

*For correspondence:
hiroki.shibuya@gu.se

**Competing interests:** The authors declare that no competing interests exist.

## Introduction

Telomeres at the ends of linear eukaryotic chromosomes are composed of tandem repeats of short G-rich DNA sequences – (TTAGGG)n in vertebrates – and sequence-specific DNA-binding proteins (*Palm and de Lange, 2008*). The telomere nucleoprotein complex has various pivotal functions such as protection of chromosome ends, lengthening of the terminal DNA, and promotion of meiotic homolog pairing (*Dilley and Greenberg, 2015*; *Shay, 2016*; *Shibuya et al., 2015*; *Shibuya et al., 2014*; *Zhang et al., 2017*). Although telomeres have ancient evolutionarily conserved roles and conserved DNA sequences, the telomeric double-strand DNA (dsDNA)-binding proteins have evolved rapidly, a phenomenon referred to as the telomere paradox (*Saint-Leandre and Levine, 2020*). Despite their low-sequence conservation, telomeric dsDNA-binding proteins in a wide variety of species – such as the fission yeast protein Taz1, the plant protein RTBP1, the protist (*Trypanosoma*) protein tbTRF, and the mammalian proteins TRF1 and TRF2 – typically have a single MYB-like DNA-binding domain (MYB) at their C-termini (known as a telobox) that directly recognizes the telomeric dsDNA in a sequence-specific manner (*Broccoli et al., 1997*; *Li et al., 2005*; *Spink et al., 2000*; *Yu et al., 2000*).

A deviation is found in budding yeast, where telomeric dsDNA is bound by the Rap1 protein with two tandem MYB domains (*Konig et al., 1996*; *Krauskopf and Blackburn, 1996*), and this deviation has been attributed to the atypical telomeric dsDNA sequence (heterogeneous and not GC-rich) in this organism (*Červenák et al., 2017*). Another deviation is found in *Caenorhabditis elegans*, which

has a typical telomeric dsDNA sequence (TTAGGC)n, but its genome does not have any TRF-like single MYB domain proteins or RAP1-like telomeric proteins (*Wicky et al., 1996*). There have been several reports showing that some transcription factors and chromatin remodelers, such as CEH-37 and HMG-5, bind to telomeric dsDNA in *C. elegans*, but deletions of these factors did not show any chromosomal defects, thus leaving the true regulator of telomeric dsDNA unidentified (*Im and Lee, 2003*; *Kim et al., 2003*; *Lanjuin et al., 2003*). It is curious why *C. elegans* lost the typical telomeric dsDNA-binding proteins while retaining the typical telomeric DNA sequence and how they maintain telomeric functions without these telomeric proteins. The identification of telomeric dsDNA-binding proteins in this organism will provide information on how general telomeric function is ensured by different telomeric dsDNA-binding proteins.

The recognition of telomeric dsDNA via dsDNA-binding proteins leads to assemblies of downstream telomere-associating proteins, thus forming the shelterin complexes (*Palm and de Lange, 2008*). An evolutionarily conserved component of the shelterin complex is the protection of telomere (POT) proteins, which directly recognize the telomeric single-strand DNA (ssDNA) through their conserved OB-fold domains. POT proteins generally act as negative regulators of telomerases through competitive binding to the telomeric ssDNA (*Kelleher et al., 2005*). Different from the dsDNA-recognition proteins, the POT proteins are well conserved, including in *C. elegans*, and it is reported that POT-1 and POT-2 (also known as CeOB2 and CeOB1) in *C. elegans* function as negative regulators of telomerase (*Raices et al., 2008*; *Shtessel et al., 2013*).

In this study, we screened for proteins that bind to POT-1 and identified two uncharacterized proteins, double-strand telomeric DNA-binding protein 1 and 2 (DTN-1 and DTN-2), in *C. elegans*. DTN-1 and DTN-2 are paralogous proteins, sharing 70% amino acid identity with each other. We performed secondary structure predictions and identified three tandem MYB domains at their N-termini, and we found that they exhibited sequence-specific dsDNA-binding activity toward telomeric sequences. DTN-1 and DTN-2 localized to telomeres both in somatic cells and germ cells from embryo to adulthood suggesting that they are constitutive telomere-binding proteins in vivo. Notably, the double knockout worm showed synthetic fertility defects that were transgenerationally progressive and were accompanied by various chromosomal defects, including chromosomal nondisjunction in meiosis, chromosomal bridges, and abnormal extensions of telomeric DNAs. Our findings suggest that DTN-1 and DTN-2 are bona fide telomeric dsDNA-binding proteins in *C. elegans* and that they are indispensable for the maintenance of germline immortality and telomere length homeostasis.

## Results

### DTN-1 and DTN-2 form complexes with POT-1 and POT-2

In order to identify novel telomeric proteins in *C. elegans*, we used a yeast two-hybrid (Y2H) approach to screen for POT-1-binding proteins from the *C. elegans* mixed-stage cDNA library, and we identified two functionally uncharacterized proteins encoded by the *R06A4.2* and *T12E12.3* genes (*Figure 1A* and *Figure 1—figure supplement 1*). These proteins, hereafter referred as double-strand telomeric DNA-binding proteins 1 and 2 (DTN-1 and DTN-2), respectively, have three putative MYB domains tandemly aligned in their N-terminal regions followed by a cluster of acidic amino acids in the middle (*Figure 1B* and *Figure 1—figure supplement 2*), which is similar to the domain configuration of the canonical c-MYB transcription factor. The POT-1-binding region (PBR) identified by the Y2H screening is located at the C-termini of these proteins (*Figure 1B* and *Figure 1—figure supplement 1*), where the amino acid sequences are highly conserved between DTN-1 and DTN-2 (80% identity). We confirmed by the Y2H analysis that both DTN-1 and DTN-2 bind to POT-1, but not POT-2 (*Figure 1C*), in a manner dependent on the C-terminal PBR (*Figure 1D*). In order to verify their in vivo interactions, we integrated three tandem FLAG tags followed by a GFP tag onto the endogenous *dtn-1* and *dtn-2* loci using CRISPR-Cas9 gene editing, and we purified the endogenous protein complex by FLAG immunoprecipitation (IP). Western blot showed the specific enrichment of the DTN-1-FLAG-GFP and DTN-2-FLAG-GFP proteins in the knock-in strain extracts, but not in wild type (N2) (*Figure 1E*). Western blot with polyclonal antibodies against POT-1 and POT-2 showed that both endogenous POT-1 and POT-2 proteins were co-precipitated with both DTN-1-FLAG-GFP and DTN-2-FLAG-GFP, proving that they form stable complexes in vivo

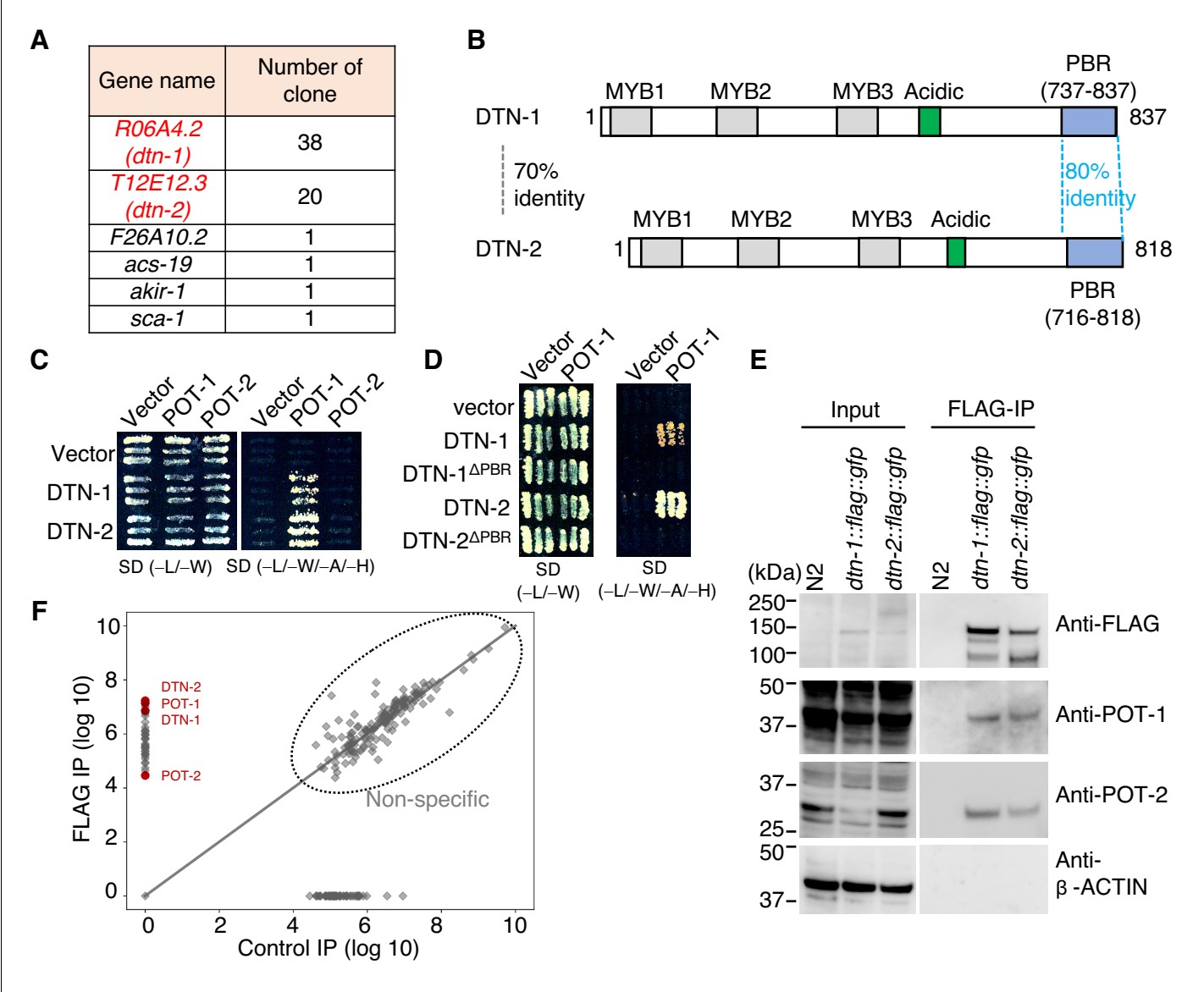

**Figure 1.** DTN-1 and DTN-2 form complexes with POT-1 and POT-2. (**A**) Genes identified in the POT-1 Y2H screening with the number of identified clones. (**B**) Schematic of the DTN-1 and DTN-2 protein sequences highlighting the MYB domains, acidic domains, and C-terminus POT-1-binding regions (PBR). The amino acid identities between the full-length sequence and the PBR of DTN-1 and DTN-2 are shown. (**C**) Y2H interactions between POT-1 and POT-2 (prey) and DTN-1 and DTN-2 (bait). AH109 yeast cells containing plasmids encoding Gal4 BD, Gal4 BD-DTN-1, Gal4 BD-DTN-2, Gal4 AD, Gal4 AD-POT-1, and Gal4 AD-POT-2 were plated on non-selective (−L/−W) and selective (−L/−W/−A/−H) plates. (**D**) Y2H interactions between POT-1 (prey) and DTN-1, DTN-1ΔPBR, DTN-2, and DTN-2 ΔPBR (bait). AH109 yeast cells containing plasmids encoding Gal4 BD, Gal4 BD-DTN-1, Gal4 BD-DTN-1ΔPBR, Gal4 BD-DTN-2, Gal4 BD-DTN-2ΔPBR, Gal4 AD, and Gal4 AD-POT-1 were plated on non-selective (−L/−W) and selective (−L/−W/−A/−H) plates. (**E**) Immunoprecipitates with the FLAG antibody from wild type (N2) and knock-in worms (*dtn-1::flag::gfp* and *dtn-2::flag::gfp*). Input and immunoprecipitates (FLAG-IP) were immunoblotted with the indicated antibodies. Note that the input blotted with anti-POT-1 was less intense compared to the FLAG-IP blotted with anti-POT-1 in order to avoid saturation of the input bands. (**F**) Quantitative mass spectrometry of immunoprecipitates with the FLAG antibody from a mixture of knock-in worms (*dtn-1::flag::gfp* and *dtn-2::flag::gfp*) (vertical axis) and control IP (horizontal axis). Combined peptide intensities are plotted for each protein. The whole protein list is provided in ***Supplementary file 1***.

The online version of this article includes the following figure supplement(s) for figure 1:

**Figure supplement 1.** Results of the POT-1 Y2H screening blue bars indicate the individual peptides identified in the Y2H screening.

**Figure supplement 2.** Structural modeling and domain conformation of DTN-1 and DTN-2.

**Figure supplement 3.** Validation of in vivo interactions between DTN-1/2, POT-1, and POT-2 immunoprecipitates with the GFP antibody from wild type (N2) and knock-in worms (*gfp::flag::pot-1* and *pot-2::gfp*).

(*Figure 1E*). Quantitative mass spectrometry analysis, which is an antibody-independent approach and is more comprehensive, also confirmed the presence of POT-1 and POT-2 in the FLAG immuno-precipitates (*Figure 1F*). The reciprocal IP experiments of GFP-POT-1 and POT-2-GFP from endogenously tagged strains also showed that both GFP-POT-1 and POT-2-GFP co-precipitated endogenous DTN-1 and DTN-2 (*Figure 1—figure supplement 3*). Together these results suggest that DTN-1 and DTN-2 are telomeric proteins in *C. elegans* that directly bind to POT-1 and indirectly bind to POT-2 in vivo.

## DTN-1 and DTN-2 bind to telomeric dsDNA

The three putative MYB domains in DTN-1 and DTN-2 are composed of three alpha helixes, which is characteristic of other MYB domains (*Figure 2A*). However, the sequence alignment of their MYB domains showed that their amino acids are highly divergent from those of known telomeric dsDNA-binding proteins found in other eukaryotes. The tryptophan residues in helices 1 and 2 (shown in the yellow rectangle in *Figure 2A*), which are known to be important for maintaining the helix-turn-helix structure and thus for the DNA-binding activity (*Zargarian et al., 1999*), are conserved in the second and third MYB domains in both DTN-1 and DTN-2. However, the basic amino acids in helix 3 (shown in the red rectangle in *Figure 2A*), which are known to make direct contact with the telomeric dsDNA (*Nishikawa et al., 2001*), are poorly conserved in DTN-1 and DTN-2. Phylogenetic analysis further confirmed that the MYB domains of DTN-1 and DTN-2 form a unique cluster that is branched from the canonical single MYB domain telomeric factors (i.e. the telobox found in TRF1/2 in mammals, RTBP1 in plants, and Taz1 in fission yeast) and the Rap1 protein in budding yeast, suggesting that DTN-1 and DTN-2 have distinct evolutionary origins from the known telomeric factors (*Figure 2B*).

To test if DTN-1 and DTN-2 have direct DNA-binding activity, we purified recombinant proteins fused with the MBP tag (*Figure 2C*) and performed an in vitro electron mobility shift assay (EMSA). Notably, both MBP-DTN-1 and MBP-DTN-2 showed robust dsDNA-binding activity toward the *C. elegans* telomeric sequence $(TTAGGC)_{15}$ but not to the scrambled sequence $(GCTGTA)_{15}$ (*Figure 2D*). The quantification of the EMSA suggested that DTN-2 ($Kd = 0.54 \pm 0.047$ μM) binds to telomeric dsDNA 1.7 times more strongly than DTN-1 ($Kd = 0.93 \pm 0.023$ μM). Both MBP-DTN-1 and MBP-DTN-2 bound only very weakly to the shorter telomeric DNA containing 1, 2, or 3 telomeric repeats, suggesting that the robust binding requires longer (more than three repeats) dsDNA (*Figure 2—figure supplement 1*) and that these proteins preferentially bind to the terminal telomere repeats rather than to the interstitial telomeric sequences under physiological conditions.

## DTN-1 and DTN-2 constitutively bind to telomeres in vivo

To determine the in vivo localization of DTN-1 and DTN-2, we analyzed the GFP signals in the knock-in worms expressing FLAG-GFP-tagged DTN-1 and DTN-2 under the control of their native promotors. Their embryos showed 18–29 punctate GFP foci specifically localized within the cell nuclei (*Figure 3A*). The average numbers of these foci per nucleus were 23 and 22 for DTN-1-FLAG-GFP and DTN-2-FLAG-GFP, respectively, which approximately corresponded to the number of telomeres in *C. elegans* (12 chromosomes and 24 telomeres). Further, the observation of pachytene oocytes in the knock-in worms' germlines showed approximately half the number of foci (*Figure 3A*, average of 12 foci per nucleus for both DTN-1 and DTN-2), which was likely due to the occurrence of meiotic homologous synapsis that reduces the apparent numbers of telomeres by half. The close observation of condensed bivalent chromosomes at the later diakinesis stage of meiosis showed eight distinct foci located at the ends of condensed chromosomes, corresponding to the telomeres of the individual chromatids (*Figure 3B*). To further confirm that these foci represent the telomeres, we performed immunostaining of the knock-in worms with a GFP antibody followed by fluorescent in situ hybridization (FISH) with a *C. elegans* telomeric DNA probe $(TTAGGC)_3$. In the embryonic nuclei, the observed GFP foci were almost completely colocalized with the telomeric FISH signals, proving that these GFP foci were bona fide telomeric signals (*Figure 3C*). In addition to embryos and germline cells, we also observed punctate GFP foci in all post-mitotic somatic nuclei in adult worms, including epidermal cells and intestinal cells (*Figure 3—figure supplement 1*). Intestinal cells in *C. elegans* become polyploid during post-embryonic development after undergoing several rounds of endomitosis (*Hedgecock and White, 1985*), and accordingly we observed numerous GFP

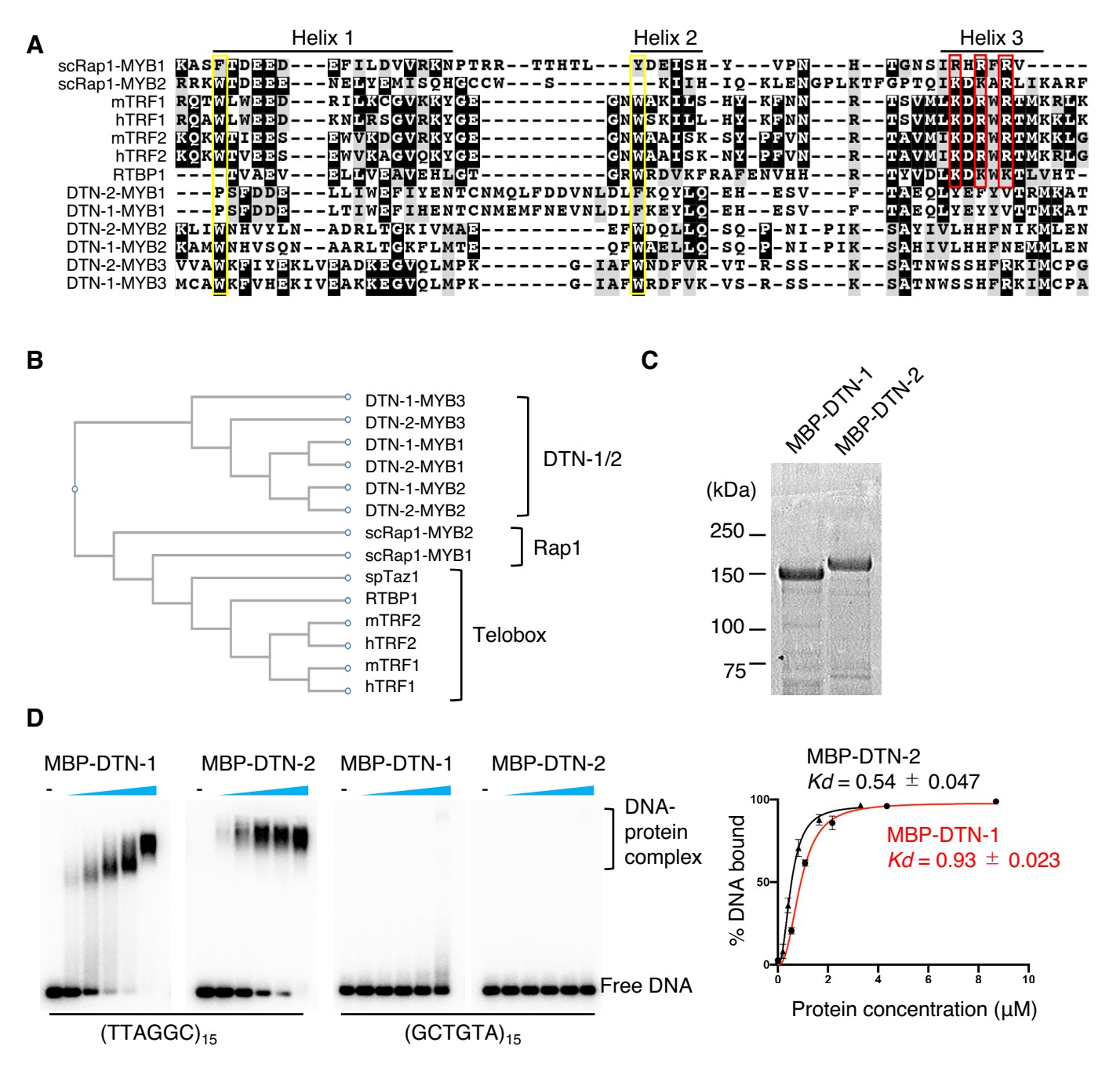

**Figure 2.** DTN-1 and DTN-2 bind to telomeric dsDNA. (**A**) Sequence alignment of the MYB domains from budding yeast Rap1 (scRap1-MYB1 and MYB2), mouse and human TRF1 and TRF2 (mTRF1, mTRF2, hTRF1, and hTRF2), rice RTBP1, and *C. elegans* DTN-1 and DTN-2. The conserved tryptophan residues required for maintaining the helix-turn-helix structure are highlighted by the yellow rectangles. Amino acids that directly contact telomeric dsDNA identified in human TRF1 protein are highlighted by the red rectangles. (**B**) Phylogenetic tree of the MYB domains. (**C**) Coomassie-stained gel of MBP-DTN-1 and MBP-DTN-2. (**D**) EMSA with increasing amounts of MBP-DTN-1 (twofold steps up to 8.7 μM) and MBP-DTN-2 (twofold steps up to 3.3 μM). The labeled DNA probes were 0.2 nM of restriction fragment containing fifteen telomere repeats (TTAGGC)$_{15}$ or scrambled repeats (GCTGTA)$_{15}$. Quantifications of the EMSA are shown in the graph to the right. Error bars are ± SD from three independent experiments. Lines are Hill curves fit to the data. The apparent affinities of MBP-DTN-1 and MBP-DTN-2 for the DNA substrates were 0.93 ± 0.023 μM and 0.54 ± 0.047 μM, respectively.

The online version of this article includes the following source data and figure supplement(s) for figure 2:

**Source data 1.** Quantifications of the EMSA from three independent experiments.

*Figure 2 continued on next page*

**Figure supplement 1.** EMSA assay with short telomeric repeats EMSA assay with increasing amounts of MBP-DTN-1 (twofold steps up to 8.8 μM) and MBP-DTN-2 (twofold steps up to 3.2 μM).

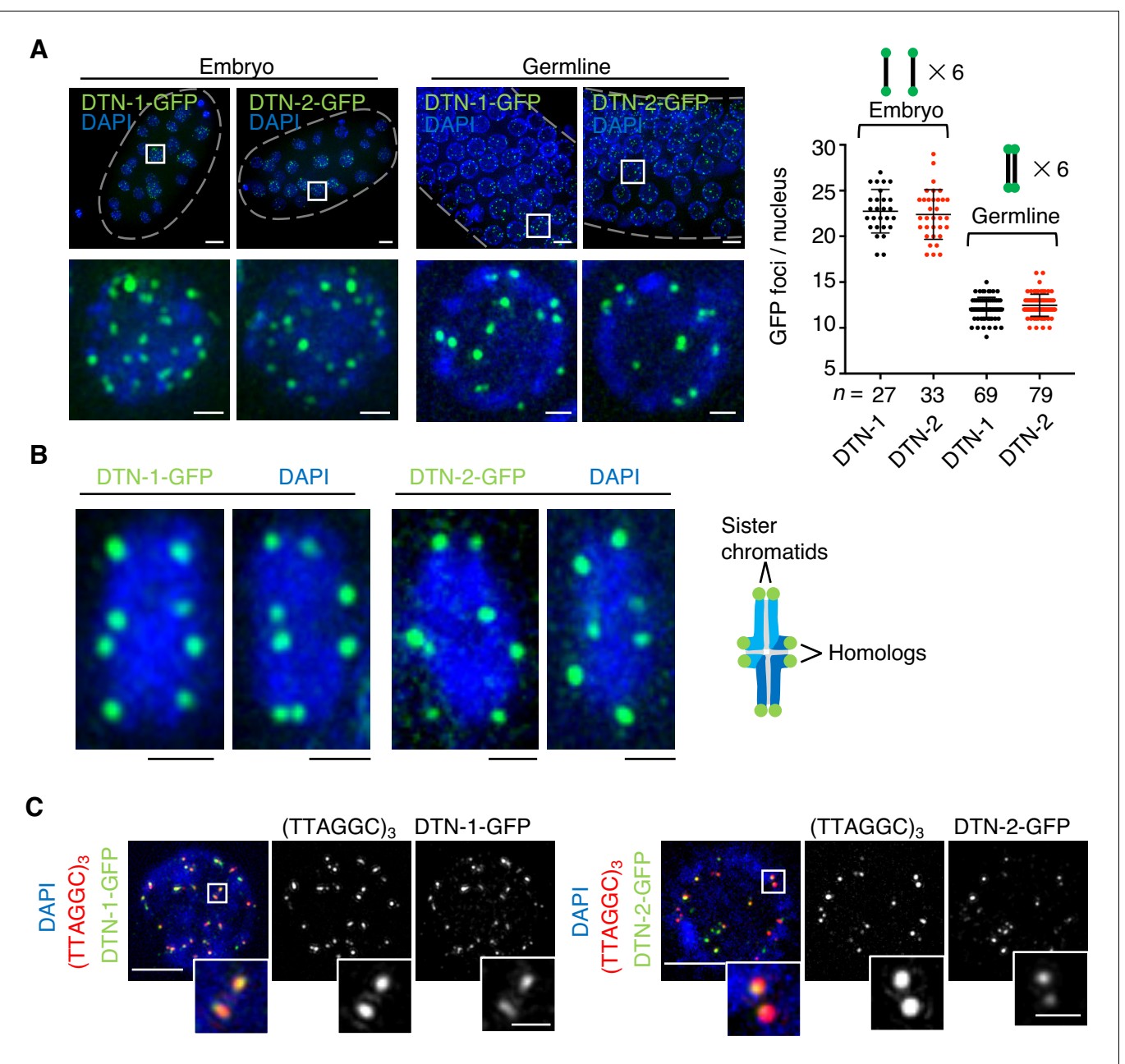

**Figure 3.** The constitutive telomeric localization of DTN-1 and DTN-2. (**A**) Embryos or germlines from knock-in worms (*dtn-1::flag::gfp* and *dtn-2::flag:: gfp*) fixed and stained with DAPI. The graph shows the number of GFP foci per nucleus. The mean value ± SD is shown. *n* shows the analyzed number of nuclei pooled from more than 10 embryos or worms. Scale bars, 5 μm or 1 μm (magnified panel). (**B**) Bivalent chromosomes from diakinesis-stage oocytes from knock-in worms (*dtn-1::flag::gfp* and *dtn-2::flag::gfp*) fixed and stained with DAPI. Scale bars, 1 μm. (**C**) Immuno-FISH of embryonic nuclei, stained with GFP antibody, hybridized with PNA probe (TTAGGC)$_3$, and stained with DAPI. Scale bars, 5 μm and 1 μm (magnified panel).

The online version of this article includes the following source data and figure supplement(s) for figure 3:

**Source data 1.** Quantifications of the number of GFP foci per nucleus.

**Figure supplement 1.** Constitutive telomeric localization of DTN-1-GFP and DTN-2-GFP.

foci in the adult intestinal nuclei (*Figure 3—figure supplement 1*) consistent with their larger telomere number compared to other somatic cells. Together, our data suggest that DTN-1 and DTN-2 localize to telomeres in both somatic and germ cells from embryo to adulthood and thus function as constitutive telomere-binding proteins in *C. elegans*.

## DTN-1 and DTN-2 are required for germline immorality

To gain insights into the functions of DTN-1 and DTN-2, we made knockout (KO) worms by deleting almost the entire coding regions of the *dtn-1* and *dtn-2* genes using CRISPR-Cas9 gene editing (*Figure 4A and B*). Western blotting using polyclonal antibodies against DTN-1 and DTN-2 confirmed that the specific bands appeared between 100 kDa and 150 kDa (close to the expected molecular weights of 95 kDa for DTN-1 and 93 kDa for DTN-2) in wild type (N2) worm extracts and that these bands completely disappeared in extracts from both corresponding KO worms (*Figure 4C*), suggesting that the protein expression was abolished in these KO worms. The western blot showed that the expression level of DTN-1 in the *dtn-2* KO worm was comparable to wild type (N2) and vice versa, suggesting that the protein stability of DTN-1 and DTN-2 is mutually independent (*Figure 4C*).

Notably, we observed no abnormalities in either of the single KO worms – they looked healthy, were maintained almost perpetually through self-fertilization under normal laboratory conditions, and had comparable numbers of progeny as wild type (N2) worms (*Figure 4D*, lane 3 and lane 7 in the graph). To investigate the possible redundancy in their functions, we crossed single KO worms to obtain double heterozygous hermaphrodites (*dtn-1$^{+/-}$; dtn-2$^{+/-}$*), which also appeared healthy and normal. From this parental strain, we isolated individual F1 progeny and performed the fertility assay. After confirming the cessation of egg laying, the genotypes of individual F1 worms were determined by single worm genotyping (*Figure 4D*). The double KO worms (*dtn-1$^{-/-}$; dtn-2$^{-/-}$*) appeared at the expected Mendelian ratio among the F1 progeny, suggesting that *dtn-1* and *dtn-2* are not essential for embryonic development (*Figure 4D*, lane nine in the graph). However, counting of brood size showed that the double KO worms exhibited severe fertility defects or were completely sterile (42% of the worms were completely sterile in the first generation). Intriguingly, retention of one intact allele of the *dtn-1* or *dtn-2* gene was sufficient to rescue the fertility defects, as shown by the normal brood sizes of *dtn-1$^{+/-}$; dtn-2$^{-/-}$* and *dtn-1$^{-/-}$; dtn-2$^{+/-}$* worms (*Figure 4D*, lane 6 and lane 8 in the graph), suggesting that these genes have redundant functions in the maintenance of fertility. We confirmed that the telomeric localization of DTN-1 and DTN-2 was mutually independent (*Figure 4—figure supplement 1*), further supporting their redundant roles at telomeres.

Even though there were only a few offspring born from the double KO worms, the continuous self-fertilization of the double KO hermaphrodites in successive generations resulted in complete sterility within a few generations, suggesting that the defect is transgenerationally progressive (*Figure 4E*). In addition to the fertility defects, we could also see a variety of morphological defects in late-generation double KO worms, such as dumpy phenotype or larval arrest (*Figure 4F*), suggesting that some somatic defects had accumulated in these worms.

## DTN-1 and DTN-2 are required for telomere length homeostasis

*C. elegans* hermaphrodites have two X chromosomes (XX), which are stably maintained during self-fertilization. Spontaneous X chromosome non-disjunction during meiosis produces male (XO) progeny, which rarely appear (~0.2%) under normal conditions (*Hodgkin et al., 1979*). During the course of our experiments, we noticed that the double KO (*dtn-1; dtn-2*) hermaphrodites produced male progeny at an abnormally high frequency (10%), suggesting that chromosomal non-disjunction occurred more frequently in meiosis in the double KO worms (*Figure 5A*). Furthermore, close inspection of the somatic nuclei revealed the prevalence of chromosomal bridges, especially in large intestinal nuclei, in the double KO worms (*Figure 5B and C*). The chromosomal bridge is a characteristic phenotype that has also been reported in mutant worms lacking genes encoding the telomerase catalytic subunit TRT-1 and in mutant worms lacking genes required for DNA replication and thus is an indication of aberrant chromosomal fusion or catenation (*Korzelius et al., 2011*; *Meier et al., 2006*). Interestingly, single or multiple telomeric FISH signals always coincided with the stretched

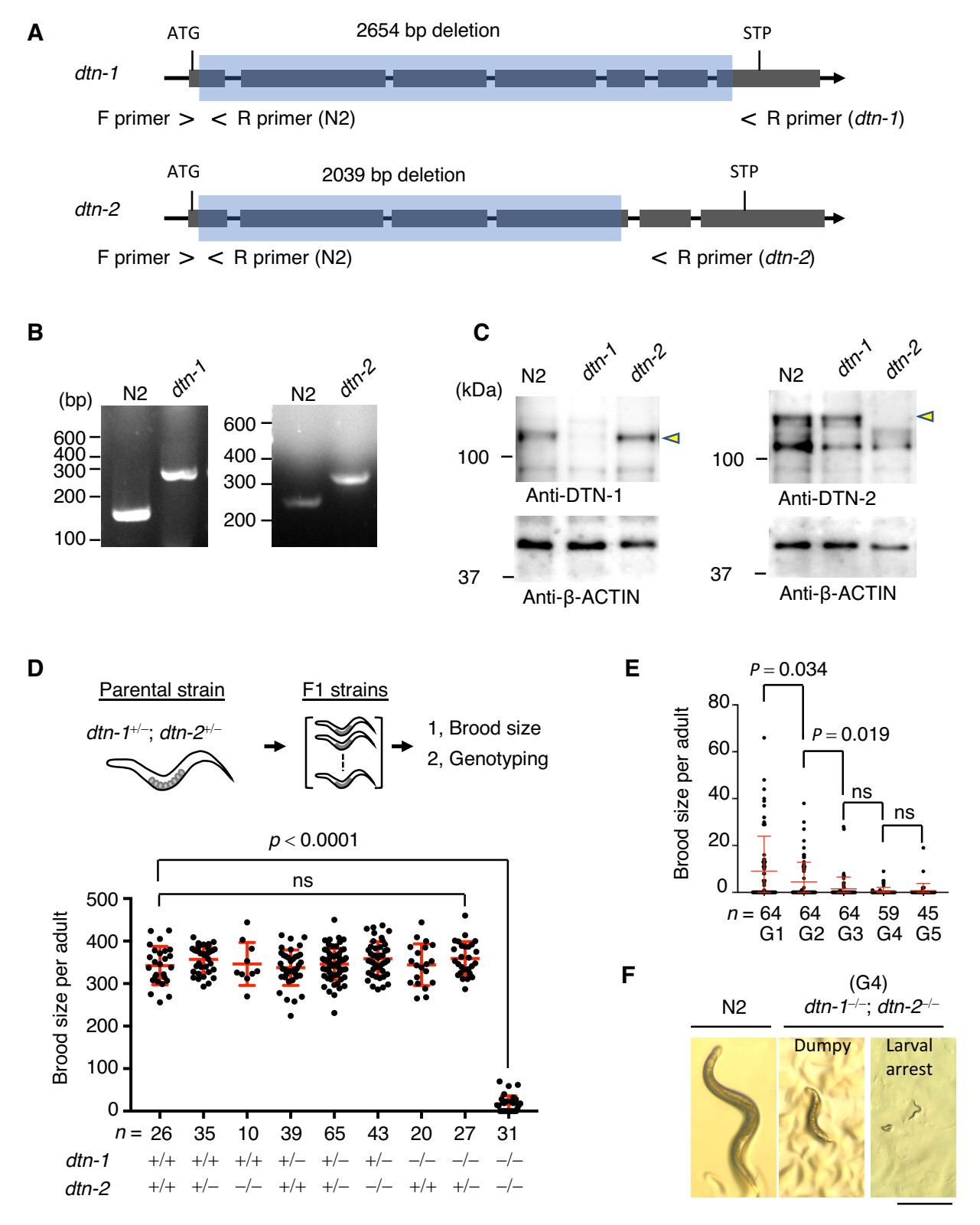

**Figure 4.** DTN-1 and DTN-2 are required for germline immortality. (**A**) Schematic of the *dtn-1* and *dtn-2* KO alleles. Exons are shown as gray rectangles with the start codon (ATG) and stop codon (STP). The deleted regions are marked by blue rectangles. Primer positions used for the genotyping are shown. (**B**) Agarose gel showing the PCR results for the *dtn-1*^+/+ (N2; 154 bp), *dtn-1*^−/− (301 bp), *dtn-2*^+/+ (N2; 267 bp), and *dtn-2*^−/− (337 bp) alleles. (**C**) Western blot with the indicated antibody for the extracts from wild type (N2) and each KO worm (*dtn-1* and *dtn-2*). Yellow arrowheads indicate the

*Figure 4 continued on next page*

*Figure 4 continued*

DTN-1 and DTN-2 proteins. (**D**) Schematic of the fertility assay. The brood size of each F1 adult worm is quantified in the graph with the genotyping results, and *n* shows the analyzed number of F1 worms for the indicated genotypes. The mean value ± SD is shown. (**E**) The brood size of *dtn-1* and *dtn-2* double KO worms self-fertilized for successive generations. *n* shows the analyzed number of worms for the indicated generations (G). The mean value ± SD is shown. (**F**) Representative morphological defects seen in *dtn-1* and *dtn-2* double KO worms at generation 4 (G4). Scale bar, 0.5 mm. Analyses were with one-way ANOVA (**D**) or two-tailed *t*-tests (**E**). ns., not significant.

The online version of this article includes the following source data and figure supplement(s) for figure 4:

**Source data 1.** The brood size quantifications.
**Figure supplement 1.** Mutually independent telomeric localization of DTN-1 and DTN-2.

DNAs between the bridging nuclei (*Figure 5D*), suggesting that these bridges likely occurred due to fusion or replication defects in their telomeric DNAs.

Because of the severe fertility defects, we could not collect large amounts of DNA samples from the double KO worms, and thus the biochemical characterization of their telomeric DNA was experimentally unfeasible. As an alternative, we performed quantitative fluorescent in situ hybridization (Q-FISH) using the telomeric probe. To eliminate artifacts caused by differences in cell cycle stage, we focused on post-mitotic somatic nuclei found in adult worms. Notably, the double KO worms had stronger telomeric FISH signals compared to wild type (N2) worms, and quantification in epidermal nuclei showed that the signal intensities in double KO worms were 5.7 times stronger than in wild type (N2) worms, suggesting that telomeric DNAs were abnormally elongated in the double KO worms (*Figure 5E*). We confirmed that the number of telomeric FISH foci in each epidermal nucleus was comparable between wild type (N2) and double KO worms, suggesting that the stronger telomere FISH signal in the double KO worms was not due to telomere fusion (*Figure 5—figure supplement 1*). Southern blot experiments showed that *dtn-1* single KO worms had abnormally elongated telomeres, while *dtn-2* single KO worms had similar or even slightly shorter telomeres compared to wild type (N2) (*Figure 5F* and *Figure 5—figure supplement 2*), which was also confirmed by Q-FISH (*Figure 5G*) suggesting that DTN-1 but not DTN-2 is responsible for the negative regulation of telomere length. Collectively, we conclude that DTN-1 and DTN-2 are redundantly required for germline immortality, while having distinct roles in the maintenance of telomere length, and if they are deleted simultaneously the worms exhibit mortal germlines accompanied by multiple chromosomal defects, including X chromosome non-disjunction in meiosis, chromosomal bridges, and hyper-elongation of their telomeric DNAs (*Figure 5H*).

## Discussion

Canonical telomeric dsDNA-binding proteins have a single MYB domain at their C-termini and are found in a number of eukaryotic species, including fission yeast, protists, plants, and mammals (*Bilaud et al., 1996*; *Broccoli et al., 1997*; *Červenák et al., 2017*; *Li et al., 2005*; *Spink et al., 2000*; *Yu et al., 2000*). Even with extensive efforts, corresponding telomeric dsDNA-binding proteins have not been identified in the nematode *C. elegans*. Conventional genetic screening for isolating mutant worms with mortal germlines successfully identified several key factors, such as MRT-2 and MRT-1 that are indispensable for telomeric DNA replication and telomerase activity respectively, but failed to identify the key telomeric dsDNA-binding proteins (*Ahmed and Hodgkin, 2000*; *Meier et al., 2009*). Using protein-interaction screening, the present study identified two non-canonical telomeric dsDNA-binding proteins, DTN-1 and DTN-2, in *C. elegans*. DTN-1 and DTN-2 are paralogous proteins and have redundant roles in the maintenance of germline immortality. Notably, even a single allele of either the *dtn-1* or *dtn-2* gene is sufficient to sustain germline immortality, which is likely to be the reason why these factors have evaded identification by conventional genetic screenings.

Structural modeling of DTN-1 and DTN-2 identified three putative MYB domains at their N-terminal regions followed by an acidic domain, which is similar to the domain configuration of the c-MYB transcription factor. The MYB domains of DTN-1 and DTN-2 are highly divergent from those of other telomeric proteins, and thus they seem to have distinct evolutionary origins. It is still unclear why these distinct telomeric proteins evolved while the telomeric DNA sequence has remained rather static (*Bilaud et al., 1996*).

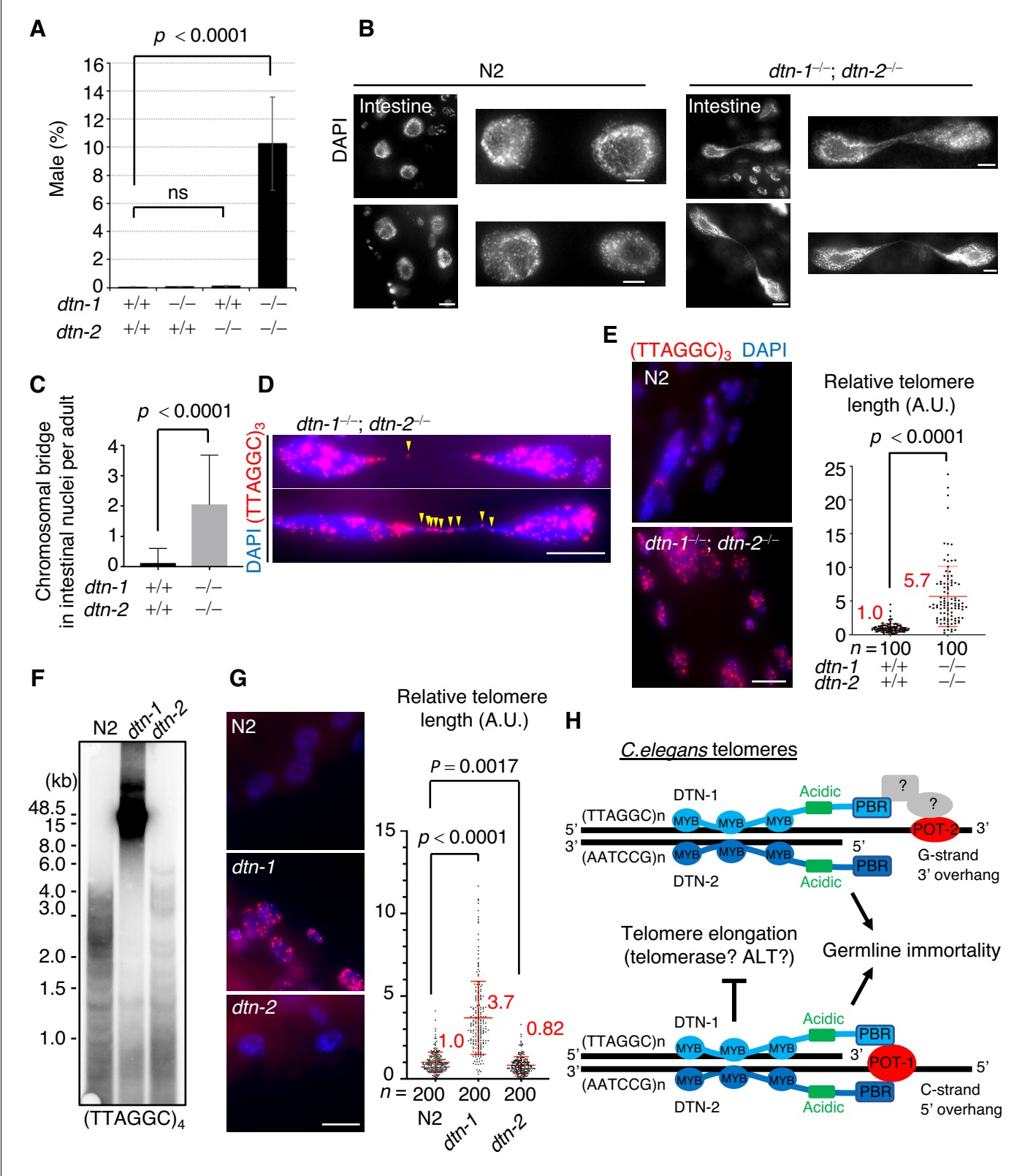

**Figure 5.** DTN-1 and DTN-2 are required for telomere length homeostasis. (**A**) The frequency of male worms among the progeny from the indicated genotypes. In total, 6551, 13270, 12317, and 1113 worms were analyzed from each genotype (*dtn-1*$^{+/+}$; *dtn-2*$^{+/+}$, *dtn-1*$^{-/-}$; *dtn-2*$^{+/+}$, *dtn-1*$^{+/+}$; *dtn-2*$^{-/-}$ and *dtn-1*$^{-/-}$; *dtn-2*$^{-/-}$). The mean value ± SD (from five biological replicates) is shown. (**B**) Intestinal nuclei from adult worms of the indicated

*Figure 5 continued on next page*

*Figure 5 continued*

genotypes stained with DAPI. Scale bars, 5 µm or 1 µm (magnified panel). (C) Quantification of the number of intestinal nuclei with chromosomal bridges per adult. A total of 17 and 18 adults were analyzed for *dtn-1*$^{+/+}$; *dtn-2*$^{+/+}$ and *dtn-1*$^{-/-}$; *dtn-2*$^{-/-}$, respectively. The mean value ± SD is shown. (D) Adult intestinal nuclei hybridized with PNA probe (TTAGGC)$_3$ and stained with DAPI. Yellow arrowheads indicate telomeric FISH signals on the bridged DNAs. Scale bar, 10 µm. (E) Adult somatic nuclei (epidermal and intestinal nuclei) hybridized with the PNA probe (TTAGGC)$_3$ and stained with DAPI. Scale bar, 5 µm. The graph shows the quantification of individual telomeric FISH signals in epidermal nuclei. The average values are normalized to that of wild type (*dtn-1*$^{+/+}$; *dtn-2*$^{+/+}$). *n* shows the analyzed number of telomeres in 10 nuclei (10 telomeres from each nuclei) pooled from five different worms. The mean value ± SD is shown. (F) Southern blot analysis of telomere length for the wild type (N2) and each single KO worm (*dtn-1* and *dtn-2*). The membrane was hybridized with DNA probes with four telomere repeats (TTAGGC)$_4$. The ladder-like hybridization signals correspond to the internal telomere sequence. (G) Adult somatic nuclei (epidermal nuclei) hybridized with the PNA probe (TTAGGC)$_3$ and stained with DAPI. Scale bar, 5 µm. The graph shows the quantification of individual telomeric FISH signals in epidermal nuclei. The average values are normalized to that of wild type (N2). *n* shows the analyzed number of telomeres in 20 nuclei (10 telomeres from each nuclei) pooled from 10 different worms. The mean value ± SD is shown. (H) Schematic summary of *C. elegans* telomere structures. Note that we did not detect any direct protein-protein interactions between DTN-1/2 and POT-2, thus they either form complexes only through DNA-mediated interactions or there are unidentified bridging proteins between them. All analyses were with two-tailed *t*-tests. ns, not significant.

The online version of this article includes the following source data and figure supplement(s) for figure 5:

**Source data 1.** Analyses of *dtn-1*, *dtn-2*, and the double KO worms.
**Figure supplement 1.** Quantification of telomeric FISH foci in epidermal cells.
**Figure supplement 2.** Quantification of Southern blot analysis.

For the recognition of dsDNA, two MYB domains function as a unit to hold the dsDNA. The canonical telomeric proteins, with single MYB domains, such as TRF1, TRF2, Taz1, and RTBP1, achieve this by forming a homodimer through their N-terminal domains (*Bianchi et al., 1997*; *Fairall et al., 2001*; *Spink et al., 2000*; *Yu et al., 2000*). In the case of the c-MYB transcription factor, two successive MYB domains (MYB2 and MYB3) within a single molecule are responsible for direct dsDNA recognition (*Sakura et al., 1989*). In the present study, we have shown that both DTN-1 and DTN-2 have robust sequence-specific dsDNA-binding activity toward the *C. elegans* telomeric sequence. It will be interesting to investigate how the three tandem MYB domains in DTN-1 and DTN-2 orchestrate substrate binding by determining their crystal structure. Structural comparison with the known telomeric proteins or with the c-MYB transcription factor might also provide insight into the evolutionary origin of these non-canonical dsDNA-binding proteins.

Telomeres are characterized by the 3′ G-rich ssDNA overhang found in almost all eukaryote species, and these overhangs are bound by the POT proteins (POT1 in human) (*Palm and de Lange, 2008*). In humans, the telomeric dsDNA-binding proteins TRF1 and TRF2 are responsible for the telomeric localization of POT1 (*Sfeir and de Lange, 2012*). TRF1 and TRF2 (as well as their accessary protein RAP1) indirectly bind to and recruit POT1 through the bridging proteins TIN2 and TPP1, and thus they form the hetero-hexameric shelterin complex TRF1-TRF2-RAP1-TIN2-TPP1-POT1 (*de Lange, 2018*). In *C. elegans*, there are both 3′ G-rich and 5′ C-rich ssDNA overhangs at terminal DNAs, which are bound by POT-2 and POT-1, respectively (*Raices et al., 2008*). We found that the C-termini of DTN-1 and DTN-2 directly bind to POT-1, and thus there seem to be no bridging proteins analogous to mammalian TIN2 and TPP1 in *C. elegans*. In this sense, the *C. elegans* shelterin complex seems to be a more simplified system, where the dsDNA recognition module is directly connected to the ssDNA recognition module. It is noteworthy that mammalian TPP1 is not merely the bridging protein required for the localization of POT1, and it also functions as a regulator of telomerase activity through direct binding to the telomerase catalytic subunit TERT (*Nandakumar et al., 2012*; *Wang et al., 2007*). DTN-1 and DTN-2 are much bigger proteins than mammalian TRF1 and TRF2, and it is possible that DTN-1 and DTN-2 have some additional functions – such as telomerase regulation – that are carried out by mammalian TPP1. Given that we could not detect any direct interaction between DTN-1/2 and POT-2, it is possible that there are uncharacterized bridging proteins linking DTN-1/2 and POT-2 in *C. elegans*, and such proteins might have an analogous function as mammalian TIN2 and TPP1. The analysis of the epistatic relationships between these proteins, as well as the screening of additional factors that bind to DTN-1 and DTN-2, should help to fully uncover the function of the *C. elegans* shelterin complex and show how telomerase is regulated in this organism.

The primary role of DTN-1 and DTN-2 in the maintenance of telomere homeostasis remains enigmatic. We have shown that the double KO worms showed signs of chromosomal abnormalities such as chromosomal non-disjunction in meiosis I (as indicated by a high incidence of male progeny) and chromosomal fusions in intestinal cells. Together with the progressive sterility phenotypes found in the double KO worms, it is speculated that these mutants are defective in the homeostasis of telomeric dsDNA, ssDNA, or both. Indeed, our data showed that the double KO and *dtn-1* single KO worms, but not *dtn-2* single KO worms, had extremely long telomeres compared to wild type (N2). These findings suggest that DTN-1 and DTN-2 have distinct roles in the maintenance of telomere length, and the role of DTN-1 seems similar to that of fission yeast Taz1, budding yeast Rap1, and mammalian TRF1, the deletions or mutations of which result in the hyper elongation of telomeric DNA (*Cooper et al., 1997*; *Krauskopf and Blackburn, 1996*; *van Steensel and de Lange, 1997*). Notably, the deletion of *pot-1*, *pot-2*, or both in *C. elegans* results in the hyper elongation of telomeres in a manner dependent on telomerase, but these worms do not exhibit chromosomal fusion or a high incidence of male progeny and are completely fertile (*Raices et al., 2008*; *Shtessel et al., 2013*). This suggests that the hyperelongated telomeres in the *dtn-1* and *dtn-2* double KO worm are less likely to be the primary reason for the observed chromosomal defect and their mortal germlines, and thus there must be some additional defects such as deprotection of telomeres and subsequent activation of the DNA-damage response pathway in the double KO worms. Further analyses will show how DTN-1 and DTN-2 protect telomeric DNAs and how evolutionarily conserved telomeric function is ensured by these distinct telomeric proteins.

# Materials and methods

**Key resources table**

| Reagent type (species) or resource | Designation | Source or reference | Identifiers | Additional information |
|---|---|---|---|---|
| Gene (*Caenorhabditis elegans*) | *R06A4.2 / dtn-1* | This paper | N/A | (cloned from a mix stage cDNA library) |
| Gene (*Caenorhabditis elegans*) | *T12E12.3 / dtn-2* | This paper | N/A | (cloned from a mix stage cDNA library) |
| Strain, strain background (*Caenorhabditis elegans*, hermaphrodite) | N2 (wild type) | Caenorhabditis Genetics Center (CGC); https://cbs.umn.edu/cgc/home | N2 | |
| Strain, strain background (*Caenorhabditis elegans*, hermaphrodite) | *dtn-1* | This paper | *Aelle; syb1925* Strain: PHX1925 | |
| Strain, strain background (*Caenorhabditis elegans*, hermaphrodite) | *dtn-2* | This paper | *Aelle; syb1886* Strain: PHX1886 | |
| Strain, strain background (*Caenorhabditis elegans*, hermaphrodite) | *dtn-1::flag::gfp* | This paper | *Aelle; syb2016* Strain: PHX2016 | |
| Strain, strain background (*Caenorhabditis elegans*, hermaphrodite) | *dtn-2::flag::gfp* | This paper | *Aelle; syb1995* Strain: PHX1995 | |

*Continued on next page*

*Continued*

| Reagent type (species) or resource | Designation | Source or reference | Identifiers | Additional information |
|---|---|---|---|---|
| Strain, strain background (*Caenorhabditis elegans*, hermaphrodite) | *dtn-1::flag::gfp; dtn-2* | This paper | *Aelle; dtn-1::flag::gfp; dtn-2* Strain: HS001 | |
| Strain, strain background (*Caenorhabditis elegans*, hermaphrodite) | *dtn-1; dtn-2::flag::gfp* | This paper | *Aelle; dtn-1; dtn-2::flag::gfp* Strain: HS002 | |
| Strain, strain background (*Caenorhabditis elegans*, hermaphrodite) | *nT1[qIs51]/dtn-2; dtn-1/dtn-1* | This paper | *Aelle; nT1[qIs51]/syb1886; syb1925/syb1925* Strain: PHX2217 | |
| Strain, strain background (*Caenorhabditis elegans*, hermaphrodite) | *gfp::flag::pot-1* | This paper | *Aelle; syb3002* Strain: PHX3002 | |
| Strain, strain background (*Caenorhabditis elegans*, hermaphrodite) | *pot-2::gfp* | This paper | *Aelle; syb889* Strain: PHX889 | |
| Antibody | anti-GFP (Rabbit polyclonal) | Invitrogen | Cat#A11122, LOT#2015993 | IF (1:1000), WB (1:1000) |
| Antibody | anti-DTN-1 (Rabbit polyclonal) | This paper | N/A | WB (1:1000) |
| Antibody | anti-DTN-2 (Rabbit polyclonal) | This paper | N/A | WB (1:1000) |
| Antibody | anti-POT-1 (Rabbit polyclonal) | This paper | N/A | WB (1:1000) |
| Antibody | anti-POT-2 (Rabbit polyclonal) | This paper | N/A | WB (1:1000) |
| Antibody | anti-β-ACTIN (Mouse monoclonal) | Sigma | Cat#A2228-200UL, LOT#067M4856V | WB (1:1000) |
| Antibody | anti-GFP (Mouse monoclonal) | Roche | Cat#11814460001, LOT#42903200 | IP |
| Antibody | Donkey Anti-Rabbit Alexa 488 | Invitrogen | Cat#A21206, LOT#1834802 | 1:1000 |
| Antibody | Peroxidase Goat Anti-Mouse IgG | Bio Rad | Cat#170–6516 | 1:1000 |
| Antibody | Peroxidase Goat Anti-Rabbit IgG | Bio Rad | Cat#170–6515 | 1:1000 |
| Recombinant DNA reagent | pMAL-c5X-*dtn-1* (plasmid) | This paper | N/A | |
| Recombinant DNA reagent | pMAL-c5X-*dtn-2* (plasmid) | This paper | N/A | |
| Recombinant DNA reagent | pET28c+-*dtn-1* (a.a. 441–837) | This paper | N/A | |
| Recombinant DNA reagent | pET28c+-*dtn-2* (a.a. 434–818) | This paper | N/A | |
| Recombinant DNA reagent | pET28c+-*pot-2* | This paper | N/A | |

*Continued on next page*

Continued

| Reagent type (species) or resource | Designation | Source or reference | Identifiers | Additional information |
|---|---|---|---|---|
| Recombinant DNA reagent | pGEX-6P-1-*pot-1* (a.a. 100–300) | This paper | N/A | |
| Recombinant DNA reagent | pB27-*pot-1* | Hybrigenics Services | N/A | |
| Recombinant DNA reagent | pP6-mix-staged *C. elegans* cDNA | Hybrigenics Services | N/A | |
| Recombinant DNA reagent | pGBKT-7-*dtn-1* | This paper | N/A | |
| Recombinant DNA reagent | pGBKT-7-*dtn-2* | This paper | N/A | |
| Recombinant DNA reagent | pGBKT-7-*dtn-1ΔPBR* (a.a. 1–736) | This paper | N/A | |
| Recombinant DNA reagent | pGBKT-7-*dtn-2ΔPBR* (a.a. 1–715) | This paper | N/A | |
| Recombinant DNA reagent | pGADT7-POT-1 | This paper | N/A | |
| Recombinant DNA reagent | pGADT7-POT-2 | This paper | N/A | |
| Sequence-based reagent | *dtn-1* genotype Common-Forward | This paper | PCR primers | 5'- CGGCAATTTGGCACGATGTT −3' |
| Sequence-based reagent | *dtn-1* genotype WT-Reverse | This paper | PCR primers | 5'- AATGACGGTCTTGACGGCTT −3' |
| Sequence-based reagent | *dtn-1* genotype KO-Reverse | This paper | PCR primers | 5'- TGGCCCAAAATCAGCCTCAA −3' |
| Sequence-based reagent | *dtn-2* genotype Common-Forward | This paper | PCR primers | 5'- TTGCGCTTTTGCTTCATCCG −3' |
| Sequence-based reagent | *dtn-2* genotype WT-Reverse | This paper | PCR primers | 5'- CTCCGCCGTAAACACAGACT −3' |
| Sequence-based reagent | *dtn-2* genotype KO-Reverse | This paper | PCR primers | 5'- GGGCACCAGAGGTAACTTCA −3' |
| Sequence-based reagent | *dtn-1::flag::gfp* genotype Common-Forward | This paper | PCR primers | 5'- GGCAACGTCGAGAACGAGAA −3' |
| Sequence-based reagent | *dtn-1::flag::gfp* genotype WT-Reverse | This paper | PCR primers | 5'- ATGACTAGGGCGAGGGGTAA −3' |
| Sequence-based reagent | *dtn-1::flag::gfp* genotype mutant-Reverse | This paper | PCR primers | 5'- CACCCTCTCCACTGACAGAAAA −3' |
| Sequence-based reagent | *dtn-2::flag::gfp* genotype Common-Forward | This paper | PCR primers | 5'- GCACAGAAGCCATCCGAAAA −3' |
| Sequence-based reagent | *dtn-2::flag::gfp* genotype WT-Reverse | This paper | PCR primers | 5'- TAGGGCTGAGGCTAAAGAATGAA −3' |
| Sequence-based reagent | *dtn-2::flag::gfp* genotype mutant-Reverse | This paper | PCR primers | 5'- TCACCCTCTCCACTGACAGA −3' |
| Sequence-based reagent | *gfp::flag::pot-1* genotype Common-Forward | This paper | PCR primers | 5'- TATGCAACGAACGAGGCTCC −3' |
| Sequence-based reagent | *gfp::flag::pot-1* genotype WT-Reverse | This paper | PCR primers | 5'- GACCCGGTACCAAATCCTGA −3' |

*Continued on next page*

*Continued*

| Reagent type (species) or resource | Designation | Source or reference | Identifiers | Additional information |
|---|---|---|---|---|
| Sequence-based reagent | *gfp::flag::pot-1* genotype mutant-Reverse | This paper | PCR primers | 5'- ATGTTGCATCACCTTCACCCT −3' |
| Sequence-based reagent | *pot-2::gfp* genotype Common-Forward | This paper | PCR primers | 5'- CGAAAACATTCGCTGAGGCT −3' |
| Sequence-based reagent | *pot-2::gfp* genotype WT-Reverse | This paper | PCR primers | 5'- GCTAGCGCCACAACCAAAC −3' |
| Sequence-based reagent | *pot-2::gfp* genotype mutant-Reverse | This paper | PCR primers | 5'- TGTTGCATCACCTTCACCCT −3' |
| Software, algorithm | SoftWoRx | GE healthcare life science | | http://www.gelifesciences.com/webapp/wcs/stores/servlet/productById/en/GELifeSciences-se/29065728 |
| Software, algorithm | CLUSTALW | https://www.genome.jp/tools-bin/clustalw | | |
| Software, algorithm | Phyre2 | http://www.sbg.bio.ic.ac.uk/~phyre2/html/page.cgi?id=index | | |
| Software, algorithm | Jpred 4 | http://www.compbio.dundee.ac.uk/jpred/ | | |

## Strains

Worms were grown at 20°C and maintained as described (*Brenner, 1974*). The following strains were used in this study: Bristol N2 wild strain, *dtn-1*(*syb1925*), *dtn-2* (*syb1886*), *dtn-1::flag::gfp* (*syb2016*), *dtn-2::flag::gfp* (*syb1995*), HS001 (*syb1925/syb1925; syb1995/syb1995*), HS002 (*syb2016/syb2016; syb1886/syb1886*), PHX2217 (*nT1[qIs51]/syb1886; syb1925/syb1925*), *gfp::flag::pot-1* (*syb3002*), *pot-2::gfp* (*syb889*). *dtn-1*(*syb1925*) was crossed with *dtn-2* (*syb1886*) to isolate double heterozygous worms. The double KO worms (*syb1925/syb1925; syb1886/syb1886*) generated from the balancer strain (PHX2217) were used in all experiments shown in *Figure 5*. All mutant alleles were generated by CRISPR-Cas9 and verified by PCR and sequencing. PCR primers used for the genotyping are listed in the key resources table.

## Homology search and structural prediction

The homology search for MYB domains and the subsequent pyrogenetic analysis were performed using the CLUSTALW program (https://www.genome.jp/tools-bin/clustalw). The presence of MYB domains in DTN-1 and DTN-2 was predicted by structural modeling using Phyre2 (http://www.sbg.bio.ic.ac.uk/~phyre2/html/page.cgi?id=index) and manual alignment. The prediction of protein secondary structure was performed using Jpred 4 (http://www.compbio.dundee.ac.uk/jpred/).

## Y2H assay

Y2H screening was performed by Hybrigenics Services, Paris, France. The coding sequence for *pot-1* was cloned into pB27 as a C-terminal fusion to LexA (LexA-*pot-1*). The construct was used as a bait to screen a random-primed *C. elegans* mixed-stage cDNA library constructed in pP6. Using a mating approach with YHGX13 and L40ΔGal4 yeast strains, 176 million clones were screened. A total of 82 positive colonies were selected on selective plates. The prey fragments of the positive clones were amplified by PCR and sequenced at their 5' and 3' junctions. The resulting sequences were used to identify the corresponding interacting proteins in the GenBank database (NCBI) using a fully automated procedure. For the yeast two-hybrid assay, *dtn-1*, *dtn-2*, *dtn1ΔPBR* (a.a. 1–736), and *dtn2ΔPBR* (a.a. 1–715) cDNAs were cloned into the pGBKT7 vector. *pot-1* and *pot-2* cDNAs were cloned into the pGADT7 vector. These bait and prey were co-transformed into the yeast strain AH109, and the positive transformants were selected on nutrition-restricted plates (SD-tryptophan-leucine-histidine-adenine).

## Recombinant protein purification

*dtn-1* and *dtn-2* cDNAs were cloned into pMAL-c5X (New England Biolabs) for expression with an N-terminal MBP tag. Constructs were expressed in BL21 (DE3) cells (Thermo Fisher Scientific) and induced with 0.4 mM IPTG for 16 hr at 15˚C. Cell disruption was achieved by sonication in extraction buffer (50 mM Tris-HCl (pH 7.5), 150 mM NaCl, 0.1% Triton X-100, and 1 mM β-mercaptoethanol), and cellular debris was removed by centrifugation at 40,000 × *g*. Fusion proteins were purified through amylose beads (NEB).

## EMSA

To prepare the DNA probes, NotI/NdeI fragments containing 15 telomere repeats (TTAGGC) or scrambled repeats (GCTGTA) were radiolabeled with $[\gamma\text{-}^{32}P]$ ATP by T4 polynucleotide kinase (New England Biolabs). For preparation of shorter DNA probes with one, two, and three repeats of telomeric DNA, the complementary oligonucleotides were annealed and radiolabeled with $[\gamma\text{-}^{32}P]$ ATP by T4 polynucleotide kinase. Proteins were mixed with 0.2 nM of labeled probes for one reaction in binding buffer (10 mM Tris-HCl (pH 7.5), 50 mM NaCl, 4% glycerol, 0.5 mM EDTA, 1 mM MgCl$_2$ 0.5 µg poly[dI-dC], 0.5 mM DL-dithiothreitol) and electrophoresed in a 0.8% agarose gel in 0.5× Tris-borate-EDTA at room temperature.

## Antibodies

The following antibodies were used: rabbit antibodies against DTN-1 (this study) 1:1000, DTN-2 (this study) 1:1000, POT-1 (this study) 1:1000, POT-2 (this study) 1:1000, and GFP (Invitrogen; A11122) 1:1000, and mouse antibody against β-ACTIN (Sigma; A2228-100UL) 1:1000 and GFP (Roche; 11814460001).

## Antibody production

cDNAs encoding the C-terminus of *dtn-1* (a.a. 441–837), the C-terminus of *dtn-2* (a.a. 434–818), and full-length *pot-2* were cloned into the pET28c+ vector (Millipore). cDNA encoding the C-terminus of *pot-1* (a.a. 100–300) was cloned into the pGEX-6P-1 vector (Addgene). The HIS- or GST-tagged recombinant proteins were expressed in BL21 (DE3) cells, solubilized in extraction buffer (600 mM NaCl and 50 mM Tris-HCl (pH 7.5)), and purified with Ni-nitrilotriacetic acid (QIAGEN) for the HIS tag or with glutathione agarose (Thermo Fisher Scientific) for the GST tag. The recombinant proteins were dialyzed into PBS and used to immunize the animals. The polyclonal antibodies were affinity purified on antigen-coupled Sepharose beads (GE Healthcare).

## Microscopy

Images were obtained on a microscope (Olympus IL-X71 Delta Vision; Applied Precision) equipped with 100× NA 1.40 and 60× NA 1.42 objectives, a camera (CoolSNAP HQ; Photometrics), and softWoRx 5.5.5 acquisition software (Delta Vision). The acquired images were processed with deconvolution (softWoRx 5.5.5) and Photoshop (Adobe).

## Histological analysis, FISH, and immuno-FISH

Age-matched hermaphrodites, 16–20 hr post-L4 larval stage, were dissected on coverslips in 20 µl of 1× egg buffer (containing 0.1% Tween-20 and 15 mM NaAzide). A SuperFrost Plus slide (Fisher) was immediately applied to the sample followed by freezing on dry ice. The coverslips were removed, and the slides were immediately placed in cold methanol for 1 min. The slides were post-fixed with 4% formaldehyde (diluted from fresh 37% formaldehyde). After washing with PBST, the slides were stained with DAPI. For immuno-FISH, the fixed slides were stained with GFP antibody and FITC-labeled secondary antibodies and fixed with 4% formaldehyde again (this step was skipped for FISH). After dehydration, the PNA-(TTAGGC)$_3$ probe was added to the slide. The slides were denatured at 85˚C for 10 min and hybridized at 37˚C for 4 hr. After sequential washing in 50% formamide/0.5×SSC (twice) and 1×SSC (twice) at 42˚C for 5 min each time, the slides were stained with DAPI.

## Southern blot

Asynchronously cultured *C. elegans* samples were collected for the genomic DNA extraction. A total of 15 µg of *C. elegans* genomic DNA were digested with HinfI (New England Biolabs) and RsaI (New England Biolabs) and separated on a 0.6% agarose gel at 8 V/cm for 3 hr. After transfer to the membrane, southern blotting was performed using the DNA probes with four repeats of telomeric DNA $(TTAGGC)_4$ radiolabeled with $[\gamma-^{32}P]$ ATP by T4 polynucleotide kinase.

## Immunoprecipitation

Mixed-stage worms were collected and suspended in IP buffer (20 mM HEPES (pH 7.0), 200 mM KCl, 5 mM $MgCl_2$, 10% glycerol, 0.1% Triton X-100, and 1 mM β-mercaptoethanol) supplemented with cOmplete Protease Inhibitor (Roche) and Phosphatase Inhibitor (Roche). After sonication, the cell extract was centrifuged at 50,000 × g for 30 min at 4°C and the supernatant was isolated. The extract was supplemented with Dynabeads protein A (Thermo Fisher Scientific) conjugated with 80 µg of antibodies or IgG as the negative control and incubated for 6 hr at 4°C. The beads were washed with high-salt buffer (20 mM HEPES (pH 7.0), 400 mM KCl, 5 mM $MgCl_2$, 10% glycerol, 0.1% Triton X-100, and 1 mM β-mercaptoethanol) supplemented with cOmplete Protease Inhibitor (Roche) and Phosphatase Inhibitor (Roche). The samples were eluted with 0.1 M glycine (pH 2.5).

## Fertility assay

Individual worms at the L4 larval stage were isolated and grown at 20°C. After reaching adulthood, the worms were transferred to a new plate every day until no eggs were laid, and viable progeny were counted approximately 24 hr after removing the parent. The parental strains, after the cessation of egg laying, were genotyped by PCR.

## Sample preparation for MS analysis

The MS protocol was largely similar to the method described in our earlier publication (*Zhang et al., 2020*). The eluted samples were reduced with DL-dithiothreitol at a final concentration of 100 mM at 60°C for 30 min and supplemented with sodium dodecyl sulfate to a 1.5% final concentration. The samples were then processed according to the filter-aided sample preparation method modified from *Wiśniewski et al., 2009*. In short, reduced samples were diluted with 500 µl of 8 M urea and 50 mM triethylammonium bicarbonate (TEAB) solution, transferred onto Nanosep 30 k Omega filters (Pall Life Sciences), and washed once with 500 µl and twice with 200 µl of 8M urea and 50 mM TEAB solution and twice with the digestion buffer (0.5% sodium deoxycholate and 50 mM TEAB). The reduced cysteine side chains were alkylated with 10 mM methyl methanethiosulfonate diluted in digestion buffer for 30 min at room temperature and the samples were then repeatedly washed with digestion buffer. Trypsin in digestion buffer was added (300 ng) and the sample was incubated at 37°C for 4 hr, then another 300 ng portion of trypsin was added and the mixture was incubated overnight. Digested peptides were collected by centrifugation, followed by a wash with 20 µl of the digestion buffer and further centrifugation. The peptide samples were treated using the HiPPR detergent removal resin kit (PN 88305, Thermo Fisher Scientific, Waltham, MA, USA) according to the manufacturer's instructions with 25 mM TEAB solution as the equilibration buffer. Sodium deoxycholate was precipitated and removed by acidification with 10% TFA and subsequent centrifugation. The supernatants were purified using Pierce peptide desalting spin columns (PN 89851, Thermo Fisher Scientific) according to the manufacturer's instructions. The purified peptide samples were dried on Speedvac and reconstituted in 15 µl of 3% acetonitrile and 0.2% formic acid for the liquid chromatography-mass spectrometry (LC-MS) analysis.

## LC-MS

LC-MS experiments were performed on an Orbitrap Fusion Lumos mass spectrometer interfaced with an Easy-nLC1200 nanoflow liquid chromatography system (both from Thermo Fisher Scientific). A total of 8 µl out of 15 µl of each peptide sample were trapped on an Acclaim Pepmap 100 C18 trap column (100 µm × 2 cm, particle size 5 µm, Thermo Fischer Scientific) and separated on an analytical column (75 µm × 35 cm) packed in-house with Reprosil-Pur C18 material (particle size 3 µm, Dr. Maisch, Germany) using a gradient with 0.2% formic acid in water as solvent A and 80% acetonitrile with 0.2% formic acid as solvent B at a flow rate of 300 nL/min. The elution profile was as

follows: 5% to 33% B in 77 min, 33% to 100% B in 3 min, and 100% B for 10 min. Precursor ion scans were performed at 120,000 target resolution with an m/z range of 375–1500 and an AGC target of 4e5. The most abundant precursors with charges 2–7 were selected for fragmentation with a maximum duty cycle of 3 s and a dynamic exclusion duration of 45 s. Precursors were isolated with a 1.0 Da window and fragmented by higher energy collision-induced dissociation at 30% collision energy with a maximum injection time of 150 ms and an AGC target 5e4, and the MS$^2$ spectra were recorded at 30,000 resolution.

### MS data analysis

Peptide and protein identification and quantification were performed using Proteome Discoverer version 2.4 (Thermo Fisher Scientific). The LC-MS files were matched against the *C. elegans* reference Uniprot database (May 2020) supplemented with common proteomic contaminants (26924 proteins in total) using Mascot 2.5.1 (Matrix Science, London, United Kingdom) as a database search engine with trypsin and one allowed missed cleavage as an enzyme rule, with the precursor tolerance of 10 ppm and fragment tolerance of 0.03 Da; methionine oxidation was set as a variable modification, and methylthiolation on cysteine was set as a fixed modification. Fixed Value PSM validator was used to assess the quality of peptide matches. Precursor ion quantification was accomplished via the Minora feature detection node in Proteome Discoverer 2.4, with the maximum peak intensity values used for quantification. Transfer of identifications between the runs was disabled. Abundance values for all unique peptides were used to calculate the protein abundances, and the intensity normalization was disabled.

### Quantification and statistical analysis

The experiments were not randomized, so no statistical method was used to predetermine sample size, and the investigators were not blinded to allocation during the experiments or to outcome assessment. Each conclusion in the manuscript was based on results that were reproduced in at least three independent experiments. Sample sizes, statistical tests, and p-values are indicated in the text, figures, and figure legends.

## Acknowledgements

We thank Marc Pilon and his lab members (University of Gothenburg) for valuable discussions and generous help with daily experiments. We thank the Proteomics Core Facility of the University of Gothenburg, in particular to Maria Segeda, for preparing the IP-MS experiment. We thank Owen R Davies (Newcastle University) for valuable discussion. This work was supported by Assar Gabrielsson's Foundation FB 17–10 (HS) and O E och Edla Johanssons vetenskapliga stiftelse 253550102 (HS).

## Additional information

### Funding

| Funder | Grant reference number | Author |
| --- | --- | --- |
| Assar Gabrielssons Foundation | FB 17-10 | Hiroki Shibuya |
| O. E. och Edla Johanssons Vetenskapliga Stiftelse | 253550102 | Hiroki Shibuya |

The funders had no role in study design, data collection and interpretation, or the decision to submit the work for publication.

### Author contributions

Hiroki Shibuya, Conceptualization, Resources, Data curation, Software, Formal analysis, Supervision, Funding acquisition, Validation, Investigation, Visualization, Methodology, Writing - original draft, Project administration, Writing - review and editing; Io Yamamoto, Data curation, Formal analysis, Validation, Investigation, Visualization, Methodology; Kexin Zhang, Data curation, Formal analysis,

Investigation; Jingjing Zhang, Data curation, Investigation; Egor Vorontsov, Validation, Investigation, Visualization, Methodology

### Author ORCIDs
Hiroki Shibuya ![ORCID] https://orcid.org/0000-0002-3400-0741

### Decision letter and Author response
Decision letter https://doi.org/10.7554/eLife.64104.sa1
Author response https://doi.org/10.7554/eLife.64104.sa2

## Additional files

### Supplementary files
• Supplementary file 1. Full list of peptides identified in control IP, DTN-1-FLAG-GFP IP, and DTN-2-FLAG-GFP IP.

• Transparent reporting form

### Data availability
All data generated or analysed during this study are included in the manuscript and supporting files.

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
