## [Decision Letter]

**Acceptance summary:**

The reviewers and editors agree that the study provides important and novel information on *C. elegans* Shelterin components. The study is well-executed, providing novel and convincing insights into telomeric binding proteins and telomere length regulation in *C. elegans*. Your thoughtful responses and the additional data addressing the initial reviewers comments are much appreciated and we are delighted to accept the revised manuscript for publication in *eLife*.

**Decision letter after peer review:**

Thank you for submitting your article "Telomeric double-strand DNA-binding proteins DTN-1 and DTN-2 ensure germline immortality in *Caenorhabditis elegans*" for consideration by *eLife*. Your article has been reviewed by three peer reviewers, and the evaluation has been overseen by a Reviewing Editor and Matt Kaeberlein as the Senior Editor. The following individual involved in review of your submission has agreed to reveal their identity: Jan Karlseder, Salk Institute (Reviewer #2).

The reviewers have discussed the reviews with one another and the Reviewing Editor has drafted this decision to help you prepare a revised submission.

Summary:

The study provides compelling insights into the function of *C. elegans* Shelterin components. The consensus reached was that the study is well-executed and that isolation of nematode double stranded telomeric binding proteins carrying myb domains is a novel result. The reviewers also noted the high quality of evidence in the form of FLAG-GFP DTN-1/2 analysis in knock-in animals. The study presents convincing biochemical analysis of telomere length regulation and stability and reports evidence of redundancy of DTN-1 and DTN-2 in terms of lethality. Overall, we feel that this is an important study that should be considered for publication in *eLife*. The reviewers made some major suggestions and comments, most of which can most likely be addressed using existing data and/or through further analysis and discussion of data already presented.

1) Please address reasons why phenotypes of *dtn-1* and *dtn-2* mutants are so different when the proteins are so similar at the amino acid level? The EMSA data in Figure 2D hints at potential differences in their behaviour. As DTN-1 concentration increases, the DTN-1-DNA complex mobility changes, suggesting that increasing copies of DTN-1 are binding to DNA. This is not the case for DTN-2 and it is curious that DTN-2 migrates aberrantly on SDS-PAGE. It would be interesting to map the DNA binding sites for both proteins and to see whether in combination, they alter DNA binding e.g. Budding yeast Rap1 (2xMyb domains) binds 13bp of DNA (3UGK) but together with Rif1/2 can selectively binds much longer fragments (Thoma lab, Cell 2013).

2) One of the interesting ideas touched upon is that *C. elegans* may not need a separate shelterin component (e.g. Tpp1) to bridge dsDNA and ssDNA binding proteins. However, this is not well developed and the discussion should be deepened and expanded.

3) The specificity of anti-POT-1 and anti-POT-2 antibodies should be described especially as Figure 1E shows both antibodies detecting multiple protein bands in the input lane. Also, the FLAG-IP should be repeated with DNase to see whether the POT-2 – DTN-1/2 co-IP is DNA mediated. This is essential to support their model in Figure 5H.

4) The authors could include oligos with single to double strand transitions (3' and 5' overhangs) to determine whether these are preferences. This could give information about the complex in regards to interactions with POT-1 and POT-2.

5) The numbers of telomeric dots in the double deletions could be analyzed better. Without access to the statistics, it seems that there are many more dots than in the wild type. This could indicate extrachromosomal telomeric DNA, or telomere fragility. Along these lines, is it possible to look at the chromosome number in the double deletion, to check whether there are fusions or fragments present? I understand that there are limited possibilities due to the mortality phenotype.

6) Reviewers suggested that it would be useful to carry out telomere Southern blot in *dtn-1/2* double KO. However, they also appreciated that collecting enough material might be difficult. They suggested Q-FISH as an alternative technology but also suggested that you might simply remove the claim that dtn-1 single mutants had similar length telomeres to *dtn-1*, *dtn-2* double mutants.

Revisions expected in follow-up work:

1) It would be better if they perform southern blots of *dtn-1*; *dtn-2* DKO mutants for their TRF changes. This would strengthen their conclusions.

2) It would be better if they show telomere lengthening/shortening by generations in single mutants DKO mutants.

3) Examination of chromosomal fusions in single mutants and DKO mutant is recommended. It would be easier to observe chromosomes in arrested oocytes than in intestines.

4) As raised above, it would be better to discuss how lengthened telomeres and chromosome bridges (probably due to fusions?) are correlated in the double mutants. This is important because it is well known that telomere shortening, rather than lengthening, causes telomere deprotection, DNA damage responses, and fusion via NHEJ.

---

## [Author Response]

Revisions for this paper:1) Please address reasons why phenotypes of dtn-1 and dtn-2 mutants are so different when the proteins are so similar at the amino acid level? The EMSA data in Figure 2D hints at potential differences in their behaviour. As DTN-1 concentration increases, the DTN-1-DNA complex mobility changes, suggesting that increasing copies of DTN-1 are binding to DNA. This is not the case for DTN-2 and it is curious that DTN-2 migrates aberrantly on SDS-PAGE. It would be interesting to map the DNA binding sites for both proteins and to see whether in combination, they alter DNA binding e.g. Budding yeast Rap1 (2xMyb domains) binds 13bp of DNA (3UGK) but together with Rif1/2 can selectively binds much longer fragments (Thoma lab, Cell 2013).

As pointed out, there is a mobility change with the increased concentration of DTN-1 but not DTN-2 in the EMSA in Figure 2D. We have repeated the experiment and confirmed the reproducibility (please find the results shown in Author response image 1 with two additional replicates). We speculate that DTN-1 but not DTN-2 undergoes some conformational changes or has a tendency to homodimerize in a concentration-dependent manner, suggesting that DTN-1 might have some unique biochemical characteristics, at least in vitro. However, this is still speculative and we want to be cautious before making any interpretations regarding this observation. We are currently working on the crystal structural analyses of DTN-1 and DTN-2 in complex with DNA substrates, and this should provide an unambiguous answer to the above issue in the future.

As we wrote in the Discussion section, we totally agree with the idea that the identification of DNA binding sites will be a critical experiment for understanding the biochemistry and biology of these MYB proteins. Making random point mutants and examining their DNA binding activities in EMSA would be an experiment we could perform at this point. However, these kinds of experiments can be highly misleading and not very conclusive because we cannot distinguish whether the mutants specifically affect DNA-binding activity or the overall conformation of the protein. To avoid misinterpretations and to unambiguously identify the DNA binding sites it is essential to get structural insights, and as noted above we are currently attempting to crystallize DTN-1 and DTN-2 in complex with DNA substrates.

We agree that the absence of bridging proteins (between DTN-1/2 and POT-1) is an interesting feature of the *C. elegans* shelterin complex. We have already discussed this point in the Discussion section as below.

“We found that the C-termini of DTN-1 and DTN-2 directly bind to POT-1, and thus there seem to be no bridging proteins analogous to mammalian TIN2 and TPP1 in *C. elegans*. […] DTN-1 and DTN-2 are much bigger proteins than mammalian TRF1 and TRF2, and it is possible that DTN-1 and DTN-2 have some additional functions – such as telomerase regulation – that are carried out by mammalian TPP1.”

However, in this discussion we did not mention the possibility of unknown bridging proteins between DTN-1/2 and POT-2. Thus we have now extended the previous discussion by adding the following text.

“Given that we could not detect any direct interaction between DTN-1/2 and POT-2, it is possible that there are uncharacterized bridging proteins linking DTN-1/2 and POT-2 in *C. elegans*, and such proteins might have an analogous function as mammalian TIN2 and TPP1. The analysis of the epistatic relationships between these proteins, as well as the screening of additional factors that bind to DTN-1 and DTN-2, should help to fully uncover the function of the *C. elegans* shelterin complex and show how telomerase is regulated in this organism.”

3) The specificity of anti-POT-1 and anti-POT-2 antibodies should be described especially as Figure 1E shows both antibodies detecting multiple protein bands in the input lane.

Our POT-1 and POT-2 antibodies are polyclonal antibodies. Thus, as is the case for most polyclonal antibodies, we see multiple bands in whole cell lysates using our antibodies. To validate the specificities of our antibodies, we have generated *gfp*::*flag*::*pot-1* and *pot-2*::*gfp* strains by tagging endogenous genes by CRISPR-Cas9.

**Validation of the POT-2 antibody.** As can be seen in Author response image 2, our POT-2 antibody detected a band around the estimated molecular weight (29 kDa) from the N2 worm extract, which disappeared in the *pot-2*::*gfp* strain extract. Instead, our POT-2 antibody detected a slower migrating band around the estimated molecular weight of POT-2-GFP (56 kDa) from the *pot-2*::*gfp* strain extract. These results suggest that our POT-2 antibody does indeed specifically detect POT-2 proteins.

**Author response image 2. respfig2:** 

**Validation of the POT-1 antibody.** Most likely because of the low protein abundance, the specific bands corresponding to POT-1 were hardly detectable in whole worm extracts using our POT-1 antibody. Furthermore, as can be seen in Author response image 3, we have blotted extracts from N2 and the *gfp*::*flag*::*pot-1* strain with the highly specific GFP antibody. However, we could not detect any specific GFP-FLAG-POT-1 bands in the corresponding INPUT lane. This is in contrast to the POT-2 case, where the POT-2-GFP band was visible in the corresponding INPUT lane. These results suggest that the amount of POT-1 protein, as well as its endogenously tagged GFP-FLAG-POT-1 protein, are low, making it difficult to detect it in whole worm extracts.However, our antibody specifically detected POT-1 when POT-1 was enriched by IP. As can be seen in the Author response image 3, we performed GFP IP using a highly specific GFP monoclonal antibody (Roche) from N2, *gfp*::*flag*::*pot-1*, and *pot-2*::*gfp* extracts. In this experiment, our POT-1 antibodies specifically recognized GFP-FLAG-POT-1, but not POT-2-GFP, after IP, thus proving the specificity of the antibody.

**Author response image 3. respfig3:** 

To further strengthen our results, we performed co-IP experiments using these newly generated strains. The GFP IPs showed that both endogenous DTN-1 and DTN-2 were co-precipitated with GFP-FLAG-POT-1 and POT-2-GFP, further strengthening our conclusion that DTN1/2, POT-1, and POT-2 form a complex in vivo. We have added these results in Figure 1—figure supplement 3B.

Also, the FLAG-IP should be repeated with DNase to see whether the POT-2 – DTN-1/2 co-IP is DNA mediated. This is essential to support their model in Figure 5H.

There are two possibilities – either there are bridging proteins between DTN-1/2 and POT-2 or there are no such bridging proteins and the DTN-1/2 and POT-2 interaction (as seen by the IPs) is mediated by DNA. In Figure 5H, we did not specify which model is true. We would like to avoid making premature conclusions and be open to both possibilities at this point. We are currently trying to identify the putative bridging proteins by POT-2 IP followed by mass spectrometry as well as by the POT-2 yeast two-hybrid screening. We hope to be able to settle this argument in future studies. To make this point clear, we have added a description to the figure legend of Figure 5H as follows.

“Schematic summary of *C. elegans* telomere structures. Note that we did not detect any direct protein-protein interactions between DTN-1/2 and POT-2, thus they either form complexes only through DNA-mediated interactions or there are unidentified bridging proteins between them.”

4) The authors could include oligos with single to double strand transitions (3' and 5' overhangs) to determine whether these are preferences. This could give information about the complex in regards to interactions with POT-1 and POT-2.

We have tried EMSA with various DNA substrates, including double-stranded DNA, 3’ G-overhang DNA, and 5’ C-overhang DNA. As can be seen in the results shown in Author response image 4, we could detect binding to all of the DNA substrates, and the quantification showed that there was no preferential binding to single-to-double-strand transitions. However, we cannot rule out the possibility that more sensitive and quantitative assays might detect subtle differences that we could not detect by EMSA. Furthermore, it is possible that the addition of POT-1 or POT-2 in the reaction could change the substrate preference. We will attempt these advanced analyses in follow-up studies. We do not wish to include these in our current manuscript in order to avoid premature conclusions at this point.

**Author response image 4. respfig4:** (**A**) EMSA with increasing amounts of MBP-DTN-1 (twofold steps up to 8.8 μM) and MBP-DTN-2 (twofold steps up to 3.2 μM). The labeled DNA probes are 0.2 nM of ligated oligonucleotides containing random dsDNA sequences (5’-GCTGTACTGGTA-3’) followed by four telomere repeat dsDNA (ds), four telomere repeat dsDNA with two telomere repeat 3’ ssDNA overhang (3’ G-overhang), and four telomere repeat dsDNA with two telomere repeat 5’ssDNA overhang (5’ C-overhang). (**B**) Quantifications of the EMSA. Error bars are ± SD from three independent experiments. (**C**) Statistical analyses of (B) with two-tailed t-tests.

5) The numbers of telomeric dots in the double deletions could be analyzed better. Without access to the statistics, it seems that there are many more dots than in the wild type. This could indicate extrachromosomal telomeric DNA, or telomere fragility. Along these lines, is it possible to look at the chromosome number in the double deletion, to check whether there are fusions or fragments present? I understand that there are limited possibilities due to the mortality phenotype.

We totally agree with your point. The signal intensity of each telomeric FISH focus will be stronger if there is a telomere fusion. To exclude this possibility, we have quantified the foci number of telomeric FISH signals in the epidermal cells (the same cells that we used for the quantification of the signal intensities in Figure 5E). This quantification showed that there was no significant difference in the telomeric foci number between N2 wild type and double KO worms. With these results, it is now more convincing that each telomere becomes longer in the double KO worms compared to the N2 wild type worms. We have added these results as Figure 5—figure supplement 1.

Furthermore, we have added the following sentence in the Results section.

“We confirmed that the number of telomeric FISH foci in each epidermal nucleus was comparable between wild type (N2) and double KO worms, suggesting that the stronger telomere FISH signal in the double KO worms was not due to telomere fusion (Figure 5—figure supplement 1).”

6) Reviewers suggested that it would be useful to carry out telomere Southern blot in dtn-1/2 double KO. However, they also appreciated that collecting enough material might be difficult. They suggested Q-FISH as an alternative technology but also suggested that you might simply remove the claim that dtn-1 single mutants had similar length telomeres to dtn-1, dtn-2 double mutants.

Both the *dtn-1* single mutant and the double mutant have elongated telomeres compared to N2. However, we agree with your point that there is no direct comparison between the *dtn-1* single mutant and the double mutants. According to your suggestion, we have removed the claim that *dtn-1* single mutants had similar telomere length as the double mutant.

Before

“Southern blot experiments showed that *dtn-1* single KO worms had abnormally elongated telomeres similar to double KO worms, while *dtn-2* single KO worms had similar or even slightly shorter telomeres compared to wild type (N2) (Figure 5F and Figure 5—figure supplement 2), which was also confirmed by Q-FISH (Figure 5G) suggesting that DTN-1 but not DTN-2 is responsible for the negative regulation of telomere length.”

After

“Southern blot experiments showed that *dtn-1* single KO worms had abnormally elongated telomeres, while *dtn-2* single KO worms had similar or even slightly shorter telomeres compared to wild type (N2) (Figure 5F and Figure 5—figure supplement 2), which was also confirmed by Q-FISH (Figure 5G) suggesting that DTN-1 but not DTN-2 is responsible for the negative regulation of telomere length.”

Revisions expected in follow-up work:1) It would be better if they perform southern blots of dtn-1; dtn-2 DKO mutants for their TRF changes. This would strengthen their conclusions.

We strongly agree with your point, but we have described the technical barrier in the Results section as below.

“Because of the severe fertility defects, we could not collect large amounts of DNA samples from the double KO worms, and thus the biochemical characterization of their telomeric DNA was experimentally unfeasible. As an alternative, we performed quantitative fluorescent in situ hybridization (Q-FISH) using the telomeric probe.”

I would like to describe this in more detail here. For the Southern blot shown in Figure 5F, we have extracted genomic DNA from four of the 10 cm dishes (which corresponds to roughly 160,000 worms). For collecting the double mutant worms, we need to collect GFP-negative worms one by one from progenies derived from the balancer strain (nT1/*dtn-2*; *dtn-1/dtn1*). It is technically not feasible to select 160,000 individual worms and thus we did not show the Southern blot of the double mutant.

2) It would be better if they show telomere lengthening/shortening by generations in single mutants DKO mutants.

We strongly agree with your point. Considering the progressive fertility defects observed in the double KO worms (Figure 4E), it is likely that telomere abnormalities might also be progressive. We will use the single telomere length analysis (STELA) method to investigate this possibility in the follow-up studies.

3) Examination of chromosomal fusions in single mutants and DKO mutant is recommended. It would be easier to observe chromosomes in arrested oocytes than in intestines.

We have quantified the foci number of telomeric FISH signals in the epidermal cells (postmitotic cells). This quantification showed that there was no significant difference in the telomeric foci number between N2 wild type and double KO worms. We have added this result as Figure 5—figure supplement 1. These results suggest that there is no significant level of chromosome fusion (end-to-end fusion), at least in epidermal cells. However, we have detected chromosomal bridges in intestinal cells (Figure 5B, C, and D) and X chromosome non-disjunction in meiosis (Figure 5A), suggesting that there must be at least some chromosomal fusion in these cells and that this might be cell-type specific. We will deepen this point by determining the telomere and chromosome number in various cell types, including oocytes, in the follow-up studies.

4) As raised above, it would be better to discuss how lengthened telomeres and chromosome bridges (probably due to fusions?) are correlated in the double mutants. This is important because it is well known that telomere shortening, rather than lengthening, causes telomere deprotection, DNA damage responses, and fusion via NHEJ.

We agree with your point. The hyper-elongated telomeres in the double mutant might not be the cause of the telomere bridges (fusions), rather the telomere elongation and deprotection might be distinct phenotypes caused by distinct mechanisms. This is supported by our findings that *dtn-1* single mutant worms, with hyper-elongated telomeres, did not show any telomere bridges (fusions). Further, previous studies also support this notion by showing that *pot-1* and *pot-2* mutant worms, which have similarly elongated telomeres, do not have any telomere deprotection phenotypes. We are sure that this is an interesting point for future study to investigate the telomere protection/deprotection states in the double mutant and to clarify the underlying mechanisms behind how *dtn-1* and *dtn-2* double KO leads to the telomere bridges (fusions) and deprotection. We have discussed this point extensively in the last part of the Discussion as follows.

“Notably, the deletion of *pot-1*, *pot-2*, or both in *C. elegans* results in the hyper elongation of telomeres in a manner dependent on telomerase, but these worms do not exhibit chromosomal fusion or a high incidence of male progeny and are completely fertile (Raices et al., 2008; Shtessel et al., 2013). […] Further analyses will show how DTN-1 and DTN-2 protect telomeric DNAs and how evolutionarily conserved telomeric function is ensured by these distinct telomeric proteins.”